# Arctigenin Reduces Myofibroblast Activities in Oral Submucous Fibrosis by LINC00974 Inhibition

**DOI:** 10.3390/ijms20061328

**Published:** 2019-03-16

**Authors:** Ching-Yeh Lin, Pei-Ling Hsieh, Yi-Wen Liao, Chih-Yu Peng, Cheng-Chia Yu, Ming-Yi Lu

**Affiliations:** 1Division of Hematology/Oncology, Department of Internal Medicine, Changhua Christian Hospital, Changhua 50006, Taiwan; 151925@cch.org.tw; 2Department of Anatomy, School of Medicine, China Medical University, Taichung 40402, Taiwan; akinosha@hotmail.com; 3School of Dentistry, Chung Shan Medical University, Taichung 40201, Taiwan; rabbity0225@gmail.com (Y.-W.L.); cyp@csmu.edu.tw (C.-Y.P.); 4Institute of Oral Sciences, Chung Shan Medical University, Taichung 40201, Taiwan; 5Department of Dentistry, Chung Shan Medical University Hospital, Taichung 40201, Taiwan

**Keywords:** oral submucous fibrosis, LINC00974, arecoline, myofibroblasts, TGF-β signaling

## Abstract

Oral submucous fibrosis (OSF) is an oral precancerous condition associated with the habit of areca nut chewing and the TGF-β pathway. Currently, there is no curative treatment to completely heal OSF, and it is imperative to alleviate patients’ symptoms and prevent it from undergoing malignant transformation. Arctigenin, a lignan extracted from *Arctium lappa*, has been reported to have a variety of pharmacological activities, including anti-fibrosis. In the present study, we examined the effect of arctigenin on the cell proliferation of buccal mucosal fibroblasts (BMFs) and fibrotic BMFs (fBMFs), followed by assessment of myofibroblast activities. We found that arctigenin was able to abolish the arecoline-induced collagen gel contractility, migration, invasion, and wound healing capacities of BMFs and downregulate the myofibroblast characteristics of fBMFs in a dose-dependent manner. Most importantly, the production of TGF-β in fBMFs was reduced after exposure to arctigenin, along with the suppression of p-Smad2, α-smooth muscle actin, and type I collagen A1. In addition, arctigenin was shown to diminish the expression of LINC00974, which has been proven to activate TGF-β/Smad signaling for oral fibrogenesis. Taken together, we demonstrated that arctigenin may act as a suitable adjunct therapy for OSF.

## 1. Introduction

Oral submucous fibrosis (OSF) is a premalignant disorder [1] characterized by chronic inflammation and progressive accumulation of collagen in the oral cavity. The common clinical symptoms include blanched mucosa and stiffness of the mouth, leading to limited food consumption, impaired speaking ability, and difficulty of maintaining oral hygiene. These symptoms cause significant social, economic, and quality-of-life burdens to patients. Moreover, the rate of mortality greatly increases once OSF progresses to neoplasia, because oral cancer is one of the most common cancers and has a fast metastasis rate [2]. Accordingly, it is crucial to explore potential therapeutic methods to attenuate OSF progression.

Epidemiological evidence has suggested that areca nut chewing is the most significant risk factor for OSF [3]. The treatment of buccal mucosal fibroblasts (BMFs) with arecoline, an alkaloid extracted from the areca nut, has been found to increase collagen synthesis and the expression of extracellular matrix-associated genes, such as tissue inhibitor of metalloproteinase-1 [4], plasminogen activator inhibitor-1 [5], and connective tissue growth factor [6]. In addition, areca nut constituents have also been shown to activate the TGF-β/p-Smad2 pathway, leading to enhanced myofibroblast activation as demonstrated by higher α-smooth muscle (α-SMA), γ-SMA, and collagen gel contraction [7,8]. It is known that TGF-β drives fibroblast–myofibroblast transdifferentiation via the induction of a contractile phenotype and up-regulation of α-SMA. Those alterations are related to the increase in collagen gene expression and are regulated by Smad proteins [9]. Consequently, a therapeutic strategy that targets the TGF-β pathway in myofibroblast transdifferentiation may serve as a promising approach.

Arctigenin is a lignan found in certain plants, such as *Arctium lappa*. Arctigenin has been reported to have numerous biological activities, including antioxidant [10], anti-inflammatory [11], and antitumor [12] properties. As for fibrosis, various studies have also demonstrated its anti-fibrotic effects. For instance, arctigenin inhibits platelet-derived growth factor-BB-activated hepatic stellate cell proliferation and arrests their cell cycle via PI3K/Akt/FOXO3a signaling [13]. In renal tubular epithelial cells, arctigenin suppresses TGF-β-induced expression of monocyte chemoattractant protein-1 (MCP-1) and the subsequent epithelial-to-mesenchymal transition (EMT) through the reactive oxygen species (ROS)-dependent ERK/NF-κB signaling pathway [14]. Moreover, arctigenin suppresses the EMT of renal tubules by reducing TGF-β1 and Smad2/3 phosphorylation, as well as up-regulating Smad7 expression [15]. These studies indicate the potential of using arctigenin as an anti-fibrosis agent via various pathways, including TGF-β. Nevertheless, its effect on the premalignant disorder OSF has yet to be evaluated.

## 2. Results

### 2.1. The Cytotoxicity of Arctigenin on Normal and Fibrotic BMFs

The chemical structure of arctigenin is shown in Figure 1A. To determine the toxic effect of arctigenin on BMFs and fibrotic BMFs (fBMFs), cell viability was examined after 48-hour treatment with various concentrations of arctigenin using MTT assay. The IC50 values of BMFs and fBMFs were 354.5 ± 23.5 μM and 133 ± 10.6 μM (Figure 1B), respectively. MTS assay also showed similar results, indicating that a lower concentration of arctigenin was sufficient to reduce the cell viability of fBMFs without damaging normal cells (Figure 1C). Thereafter, arctigenin used in the following experiments was in this concentration range to examine whether it exhibited anti-fibrotic property.

### 2.2. Arctigenin Prevents the Arecoline-Associated Myofibroblastic Transdifferentiation of BMFs

Arecoline (20 μg/mL) has been used to induce myofibroblast activation in human primary BMFs with increased expression of α-SMA and other fibrogenic genes, such as vimentin and type I collagen [16]. In order to test whether arctigenin was able to impede the effect of arecoline on BMFs, we examined a variety of phenotypic analyses of myofibroblast activities. Our results suggested that the arecoline-increased collagen gel contractility (Figure 2A) and migration ability (Figure 2B) was suppressed by arctigenin in a concentration-dependent manner. Moreover, the arecoline-induced enhancement of cell motility was avoided in the presence of arctigenin as assessed by invasion (Figure 3A) and wound healing (Figure 3B) assays. These findings indicate that arctigenin may possess the potential to obviate the excessive activation of myofibroblasts caused by areca nut.

### 2.3. Arctigenin Ameliorates the Dysregulated Myofibroblast Activities of fBMFs

It has long been known that TGF-β initiates the inflammatory response by recruiting fibroblasts to the site of injury during the early phases of tissue recovery [17], and the invasive fibroblast phenotype is essential for severe fibrogenesis [18]. Consequently, we sought to evaluate the anti-fibrotic effect of arctigenin in fBMFs. We found that the fBMFs displayed a significant reduction of cell contraction capability at 5–20 μM of arctigenin (Figure 4A), suggesting that arctigenin may be capable of relieving oral rigidity without causing damage to the normal BMFs. Moreover, we observed that cell migration was conspicuously downregulated in response to the increased concentration of arctigenin using a Transwell system (Figure 4B). Similarly, there was a marked reduction in the invasion (Figure 5A) and wound healing (Figure 5B) capacities as the concentration of arctigenin increased. Altogether, we demonstrated that arctigenin inhibits these upregulated myofibroblast activities.

### 2.4. Arctigenin Downregulates the TGF-β/p-Smad2 Pathway along with the Expression of Other Fibrogenic Markers

The TGF-β/p-Smad2 signaling pathway is implicated in oral fibrogenesis. As shown in Figure 6A, arctigenin significantly mitigates the secretion of TGF-β of two fBMFs strains in a dose-dependent fashion. In accordance with this finding, the protein expression level of phosphorylated Smad2 was downregulated, as well as myofibroblast marker α-SMA, extracellular cell matrix (ECM) molecules, and type I collagen A1, (Col1a1), following arctigenin administration (Figure 6B). Our recent work has demonstrated that long non-coding RNA LINC00974 activates TGF-β/Smad signaling to promote oral fibrogenesis [19]. We found that the TGF-β production and phosphorylated Smad2 expression were repressed in the LINC00974-inhibited myofibroblasts [19]. Arctigenin reduced the expression of Linc00974 in fBMFs (Figure 6C). In the current study, we showed that the expression of LINC00974 was dose-dependently suppressed by the administration of arctigenin using qRT-PCR analysis (Figure 6D). Collectively, our results suggest that arctigenin has anti-fibrotic effects on fBMFs via TGF-β/Smad signaling mediated by LINC00974.

## 3. Discussion

Over the past few years, growing attention has been paid to the use of natural products against a variety of diseases, including fibrosis and cancers [20,21]. Our group has also demonstrated the effect of various natural compounds on the suppression of oral cancer or OSF via the reduction of inflammation or mesenchymal transdifferentiation [22,23,24,25]. Current evidence has suggested that TGF-β signaling is the key pathway to trigger the manifestation of OSF [7,8]. It has been shown that thrombin released from microtrauma during areca nut chewing was able to induce αvβ1, αvβ3, and αvβ5 integrins-mediated TGF-β1 activation, leading to production of connective tissue growth factor (CTGF) in human BMFs [26]. Arecoline-induced mitochondrial ROS also plays a pivotal role in the activation of latent TGFβ1 and the subsequent increase of CTGF and early growth response-1 in BMFs [27]. By using epigallocatechin-3-gallate (a type of catechin), the arecoline-associated TGFβ1 activation and the following events were suppressed [26,27]. Our previous study demonstrated that downregulation of arecoline-increased TGF-β signaling by a natural compound glabridin successfully reduced myofibroblast characteristics [28]. In accordance with these studies, we showed that arctigenin also exhibits the potential to repress arecoline-stimulated TGF-β/Smad2 signaling and decrease myofibroblast activities.

Arctigenin has been proven to inhibit the proliferation of activated hepatic stellate cells via G0/G1 cell cycle arrest. Their results show that arctigenin elevates the expression of p27(Kip1) protein through inhibition of Akt and improvement of FOXO3a, resulting in the inhibition of CDK2 kinase activity for liver fibrosis [13]. In renal interstitial fibrosis, arctigenin also displayed the capacity to suppress TGF-β1-asscociated changes. It has been revealed that arctigenin inhibits the ROS/ERK1/2/NF-κB pathway to decrease MCP-1 upregulation and TGF-β1-induced EMT-like alteration in renal tubular epithelial cells [14]. Likewise, arctigenin administration dramatically decreases macrophage infiltration and pro-inflammatory cytokine production in the obstructed kidneys. Moreover, arctigenin inhibits EMT of renal tubules by reducing TGF-β1/Smad2/3 phosphorylation and nuclear translocation [15]. Apart from renal fibrosis, our results demonstrate that arctigenin is a promising anti-fibrosis agent for OSF. In the present study, we showed that the phenotypical behaviors of myofibroblasts were all mitigated. Most importantly, our findings suggest that the administration of arctigenin is capable of interfering the arecoline-induced activation of myofibroblasts in BMFs. We found that arctigenin diminishes TGF-β1/Smad2 signaling and other genes implicated in fibrosis. Furthermore, our recent study has found that TGF-β-stimulated myofibroblast activities were mediated by LINC00974 in fBMFs [19]. As a continuation of our previous work, we examined whether treatment with arctigenin could affect the expression of LINC00974 as well. As expected, we showed that the expression level of LINC00974 was inhibited in fBMFs following arctigenin treatment. Our results demonstrate that arctigenin possesses the ability to modulate non-coding RNAs, in addition to its anti-inflammation or anti-oxidative properties for fibrosis treatment. Moreover, LINC00974 also has been considered to be an oncogenic factor, It is, therefore, worthwhile to examine whether arctigenin exhibits an anti-cancer effect via regulation of LINC00974.

In summary, our data suggest that arctigenin is a suitable anti-fibrosis agent to prevent areca nut-promoted myofibroblast activation or to diminish myofibroblast characteristics via the TGF-β1/Smad2 pathway mediated by LINC00974 in oral fibrogenesis.

## 4. Materials and Methods

### 4.1. Chemicals

Arctigenin was purchased from Sigma-Aldrich Chemical Co. (St. Louis, MO, USA) and diluted in culture medium to the appropriate final concentrations prior to use. Collagen solution from bovine skin for collagen gel contraction assay was also obtained from Sigma-Aldrich.

### 4.2. Tissue Acquisition and Cell Culture

Buccal mucosa fibroblasts (BMFs) and fibrotic buccal mucosa fibroblasts (fBMFs) were retrieved from OSF tissues or the normal counterparts of patients recruited in the Oral Medicine Center (Chung Shan Medical University Hospital, Taichung, Taiwan) with informed consent. The protocol in this study was approved by the Institutional Review Board of Chung Shan Medical University Hospital (CSMUH No: CS17137; 20/08/2018). Normal BMFs and fBMFs were cultured as previously described [22]. Cell cultures between the third and eighth passages were used in this study.

### 4.3. Cell Viability Assay

To determine the cytotoxic effect of arctigenin on BMFs and fBMFs, cells were seeded at 1 × 10^4^ cells/well in the presence of various concentrations of arctigenin for 48 h. After treatment of MTT (3-(4,5dimethylthiazol-2-yl)-2,5-diphenyl tetrazolium bromide), the formazan crystals of viable cells were dissolved in DMSO. The resulting colored solution was evaluated spectrophotometrically at 570 nm. The data were presented as a percentage of DMSO [25]. As for MTS assay, MTS solution containing PES was added to each well after incubation with arctigenin. Absorbance at 490 nm was used to analyze the viable cells. IC50 values were calculated by GraFit software (Erithacus Software Ltd., West Sussex, UK).

### 4.4. Collagen Gel Contraction Assay

The biological activity of myofibroblasts was examined by collagen contraction assay (Sigma-Aldrich Chemical Co.). First, 2 × 105 cells were suspended in collagen solution (2 mg/mL), and the cell–collagen mixtures were loaded into 24-well plates until polymerization. To initiate contraction, collagen gels were gently dissociated from the sides of the well with a 200 μL pipet tip. Gels were further incubated in 0.5 mL MEMα medium for 48 h. The changes in collagen gel size (contraction index) were photographed by a digital camera at a fixed distance above the wells and quantified by ImageJ software [29].

### 4.5. Migration and Invasion Assessments

A 24-well plate Transwell^®^ system with a polycarbonate filter membrane of 8-μm pore size (Corning, Corning, NY, USA) was utilized to carry out the migration and invasion assays. For examination of invasion ability, the membrane was coated with Matrigel. The cells were added to the top compartment at a density of 1 × 10^5^ cells in 250 μL serum-free medium, and medium supplemented with 10% FBS was used as a chemoattractant in the lower chamber. After 48 h of incubation, the filter membrane was stained with 0.1% crystal violet and then visualized and counted from 5 different fields of 100-fold magnification under an inverted microscope [30].

### 4.6. Wound Healing Assay

Cells were seeded into 6-well plates and allowed to grow to around 80% confluence as a monolayer. Wound gaps were created by a sterile 200 μL plastic pipette tip. Cell movement toward the center of the wound area was photographed at 0 and 48 h under a microscope [31]. The evaluation of wound closure areas was quantified by ImageJ software (Image J, NIH, Bethesda, MD, USA).

### 4.7. Quantitative Real-Time PCR (qRT-PCR)

Total RNAs were isolated from cells using Trizol reagent according to the manufacturer’s protocol (Invitrogen Life Technologies, Carlsbad, CA, USA). qRT–PCRs of mRNAs were reverse-transcribed using the Superscript III first-strand synthesis system for RT–PCR (Invitrogen Life Technologies). SYBR green-based qRT-PCR reactions on resulting cDNAs were conducted on an ABI StepOne™ Real-Time PCR Systems (Applied Biosystems) [32].

### 4.8. Western Blot Analysis

Cells were lysed in NP-40 buffer and protein concentration was measured by a BCA protein assay kit (Thermo Fisher Scientific, Rockford, IL, USA). The samples were separated by 10% SDS-PAGE and wet-transferred to a PVDF membrane (Millipore, Billerca, MA, USA). The primary antibodies against COL1A1 (1:1000), α-SMA (1:1000), phospho-Smad2, and Smad2 (1:1000) were purchased from Santa Cruz Biotechnology, Inc. (Santa Cruz, CA, USA). GAPDH (GeneTex, Irvine, CA, USA; 1:5000) was used as the internal control. After incubation with the corresponding secondary antibodies, the immunoreactive bands were developed using an ECL-plus chemiluminescence substrate (Perkin-Elmer, Waltham, MA, USA) and captured by a LAS-1000 plus Luminescent Image Analyzer (GE Healthcare, Piscataway, NJ, USA) [33].

### 4.9. Statistical Analysis

SPSS Statistics version 13.0 was used for statistical analysis. Student’s t test or ANOVA analysis with post-hoc Tukey HSD was used to determine the statistical significance of the differences among the experimental groups; *p* values less than 0.05 were considered statistically significant.

## 5. Conclusions

In the current study, we aimed to investigate the protective effect of arctigenin against fibrogenesis and unveil the underlying mechanism. We conducted a number of phenotypic analyses, including collagen gel contraction, migration, invasion, and wound healing assays. Apart from that, we examined the expression of fibrogenic markers in response to arctigenin treatment in order to clarify whether arctigenin is a desirable agent for OSF.

## Figures and Tables

**Figure 1 ijms-20-01328-f001:**
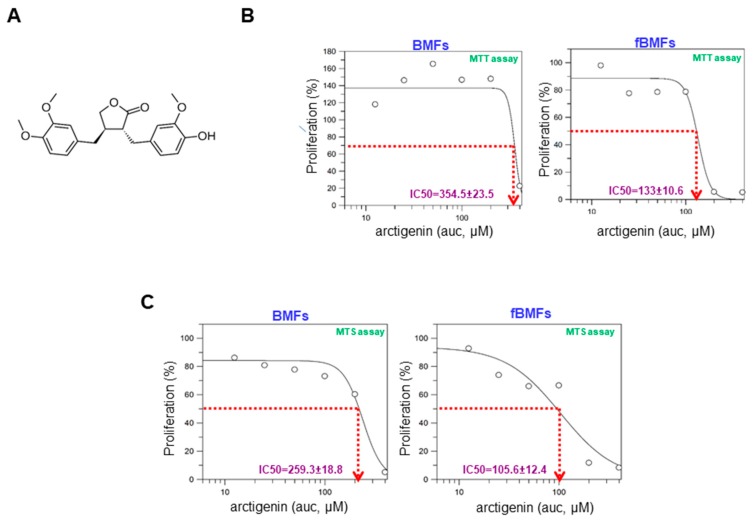
The cytotoxic effect of arctigenin on the viability of buccal mucosal fibroblasts (BMFs) and fibrotic BMFs (fBMFs). (**A**) The chemical structure of arctigenin; (**B**) MTT and (**C**) MTS assays were utilized to examine cell survival/proliferation in response to arctigenin. Normal BMFs and fBMFs were seeded at 1 × 10^4^ cells/well and treated with the indicated concentration of arctigenin for 2 days (five replicates for each concentration). The IC50 values were calculated by GraFit software.

**Figure 2 ijms-20-01328-f002:**
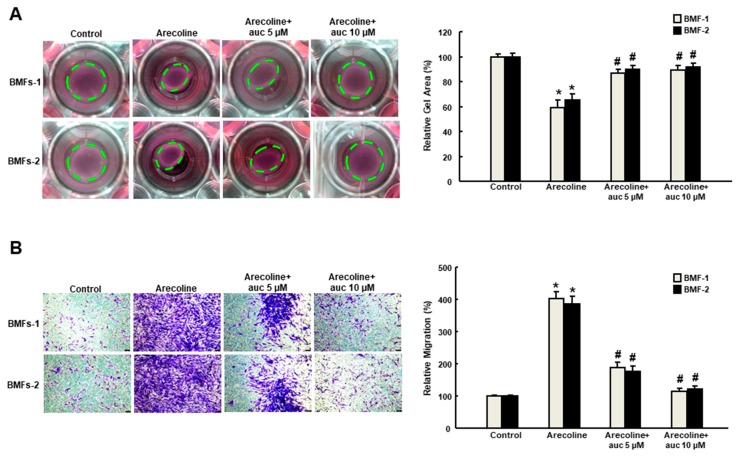
Arctigenin inhibits the arecoline-induced collagen contraction and migration capacities of BMF. (**A**) Arecoline-treated BMFs with or without arctigenin were subjected to collagen gel contraction assay (three replicates for each concentration). The gel areas (dotted circles) were calculated using ImageJ software. Representative images are presented; (**B**) Cell migration was measured 48 hours post-arctigenin exposure in arecoline-stimulated BMFs. * *p* < 0.05 as compared with the non-arctigenin-treated group; # *p* < 0.05 as compared with the arecoline group. Magnification, ×100.

**Figure 3 ijms-20-01328-f003:**
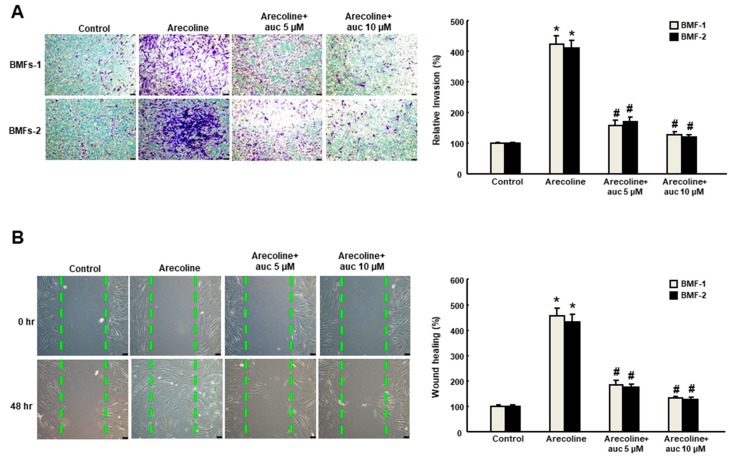
Arctigenin downregulates the arecoline-stimulated invasion and wound healing activities of BMF. Arecoline-treated BMF were subjected to (**A**) invasion or (**B**) wound healing assays in the presence of various concentrations of arctigenin. The experiments were repeated three times, and representative results are shown. The results are means ± SD. * *p* < 0.05 as compared with the non-arctigenin-treated group. # *p* < 0.05 as compared with the arecoline group. Magnification, ×100. The green dot line define the areas of wound healing cells.

**Figure 4 ijms-20-01328-f004:**
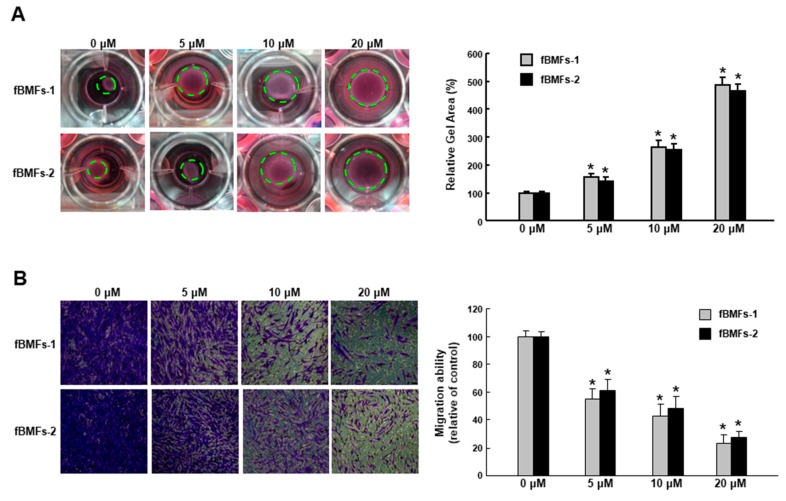
Arctigenin suppresses the collagen gel contraction and invasion capabilities of fBMFs. Myofibroblast activities were determined by (**A**) collagen gel contraction and (**B**) migration in fBMFs treated with various concentrations of arctigenin. The experiments were repeated three times, and representative results are shown. The results are means ± SD. * *p* < 0.05 as compared with the non-arctigenin-treated group. Magnification, ×200.

**Figure 5 ijms-20-01328-f005:**
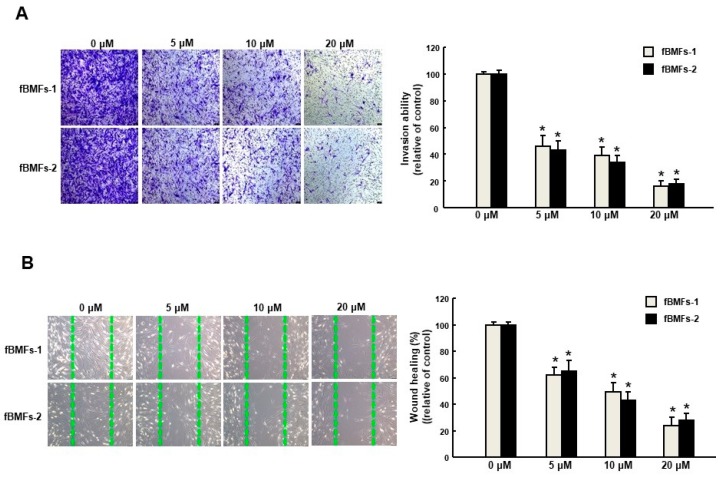
Arctigenin suppresses the myofibroblast activity of fBMFs. (**A**) Invasion and (**B**) wound healing in fBMFs treated with various concentrations of arctigenin. The experiments were repeated three times, and representative results are shown. The results are means ± SD. * *p* < 0.05 as compared with the non-arctigenin-treated group. Magnification, ×100.

**Figure 6 ijms-20-01328-f006:**
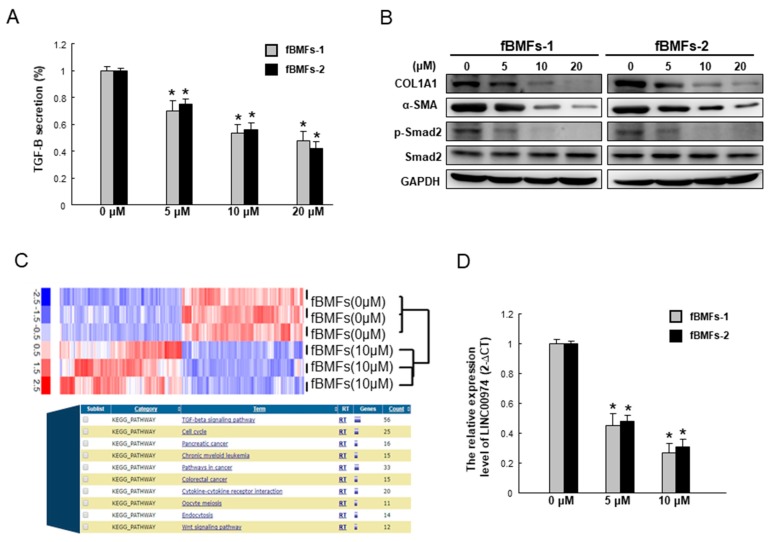
Arctigenin represses the expression of myofibroblast markers and the TGF-β/Smad2 signaling of fBMFs by targeting LINC00974. (**A**) Arctigenin dose-dependently downregulated the secretion of TGF-β1 in two fBMFs; (**B**) The protein expression levels of myofibroblast markers, including COL1A1 and α-SMA, as well as phosphor-Smad2 were reduced after arctigenin treatment in a dose-dependent manner; (**C**) The indicated lncRNAs’ expression levels in the arctigenin-treated fBMFs were analyzed by a high-throughput RNA sequencing approach and bioinformatics analysis; (**D**) qPCR analysis was applied to analyze the relative LINC00974 expression level in arctigenin-treated fBMFs. * *p* < 0.05 as compared with the non-arctigenin-treated group.

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
