# Peer review of "Arctigenin Reduces Myofibroblast Activities in Oral Submucous Fibrosis by LINC00974 Inhibition"

_ijms, 2019, doi:10.3390/ijms20061328_

Round 1

Reviewer 1 Report

the ms is ready for publication

Reviewer 2 Report

No comments

This manuscript is a resubmission of an earlier submission. The following is a list of the peer review reports and author responses from that submission.

Round 1

Reviewer 1 Report

In this manuscript titled “Arctigenin reduced myofibroblasts activities in oral submucous fibrosis by LINC00974 inhibition”, the authors have examined the antifibrotic properties of arctigenin using fibrotic buccal mucosal fibroblasts to evaluate if arctigenin may be a suitable adjunct therapy for oral submucous fibrosis.

Following are my concerns if they can be answered:

(1)   Page 2, line 66………..change “was” to “is”.

(2)   Page 2, line 67…………..why was the effect of arctigenin on cell viability of BMFs and fBMFs determined after 48 hours? Did you try checking viability after 24 or 72 hours?

This is very important because the concentrations of arctigenin used in subsequent experiments was in the concentration range of IC50 values obtained after 48 hour incubation of cells with arctigenin.

(3)   All experiments were performed after 48 hour incubation of cells with arctigenin, but for migration and invasion assays were done after 24 hr incubation. Can you justify this?

(4)   Page 6, line 129…….Place a period after “(Fig. 6B)”

(5)   Page 7, line 180…….change “possess” to “possesses”

Author Response

Reviewer 1 Comments and Suggestions for Authors In this manuscript titled “Arctigenin reduced myofibroblasts activities in oral submucous fibrosis by LINC00974 inhibition”, the authors have examined the antifibrotic properties of arctigenin using fibrotic buccal mucosal fibroblasts to evaluate if arctigenin may be a suitable adjunct therapy for oral submucous fibrosis. 

Response: We thank the reviewer for acknowledging our work, and understanding the considerable quantity of data we present herein. 

Following are my concerns if they can be answered: 

(1) Page 2, line 66………..change “was” to “is”. 

Response: Thanks for your comment. The typo is corrected as your kind advices. Please find the revised version. 

(2) Page 2, line 67…………..why was the effect of arctigenin on cell viability of BMFs and fBMFs determined after 48 hours? Did you try checking viability after 24 or 72 hours? This is very important because the concentrations of arctigenin used in subsequent experiments was in the concentration range of IC50 values obtained after 48 hour incubation of cells with arctigenin. Response: Thanks for your comment. We have examined the viability after 24 hours at the beginning (in the figure shown below and the attached file), but the IC50 value in BMFs was hard to determine even if high concentration was used. Therefore, we extended the treatment time to 48 hours and were able to better calculate the IC50 using the GraFit software. The following myofibroblast phenotypes including migration, invasion, and wound healing approaches were examined by 48 hour. As we have identified the IC50 at 48 hours, we did not test the cell viability after 72 hours. 

(3) All experiments were performed after 48 hour incubation of cells with arctigenin, but for migration and invasion assays were done after 24 hr incubation. Can you justify this?

Response: Thank you for pointing out our mistake. The description of migration and invasion assays has been corrected. We are sorry for the confusion again.

(4) Page 6, line 129…….Place a period after “(Fig. 6B)” 

Response: Thanks for your comment. The typo is corrected as your kind advices. Please find the revised version.

(5) Page 7, line 180…….change “possess” to “possesses” 

Response: Thanks for your comment. The typo is corrected as your kind advices. Please find the revised version.

Reviewer 2 Report

the ms could represent an important contribution, but before acceptance some details should be clarified.

why the authors decide to test the toxicity only at 48? what about at 24h?

MTT is not enough to understant cell viability. it is mandatory to use other cell viability tests in order to clarify the impact on cellular status.

it is not clear the timing for migration/invasion.

it is not clear the notion of fibrotic buccal mucosa fibroblasts. how the authors can verify this statement?

for wound healing assay, the authors provide little information on the image analysis for the scratch wound assays.Some detail would be useful.

Author Response

Reviewer 2

Comments and Suggestions for Authors

the ms could represent an important contribution, but before acceptance some details should be clarified.

Response: We thank the reviewer for acknowledging our work, and understanding the considerable quantity of data we present herein.

(1) why the authors decide to test the toxicity only at 48? what about at 24h?

Response: Thanks for your comment. We did examine the cell viability after 24 hours (in the figure shown below and the attached file), but it was hard to determine the IC50 value even if high concentration was used. As a result, we extended the treatment time to 48 hours and were able to calculate the IC50 using the GraFit software. The following myofibroblast phenotypes including migration, invasion, and wound healing approaches were examined by 48 hour.

(2) MTT is not enough to understant cell viability. it is mandatory to use other cell viability tests in order to clarify the impact on cellular status.

Response: Thank you for your comment. We have performed the 3-(4,5-dimethylthiazol-2-yl)-5-(3-carboxymethoxyphenyl)-2-(4-sulfophenyl)-2H-tetrazolium inner salt (MTS) assay (shown below and the attached file) to evaluate the cell viability. The results are consistent with our data of MTT assay that showed BMFs required higher concentration of arctigenin to cause cytotoxic effect.

(3) it is not clear the timing for migration/invasion.

Response: We thank you for the sound advice. These descriptions have been added in the revised text.

(4) it is not clear the notion of fibrotic buccal mucosa fibroblasts. how the authors can verify this statement?

Response: As mentioned in material and methods section, fibrotic buccal mucosa fibroblasts (fBMFs) were retrieved from OSF tissues. We have successful cultivated and identified enhanced myofibroblasts markers and phenotypes in fBMFs (Sci Rep. 2016; 6: 37004.)

(5) for wound healing assay, the authors provide little information on the image analysis for the scratch wound assays. Some detail would be useful.

Response: We thank you for the sound advice. These descriptions have been added in the revised text.

Round 2

Reviewer 2 Report

the authords did not reply to main question about the use of other tests that MTT assay. They provided only a easy description of the test in the rebuttal letter.